https://doi.org/10.1038/s41467-019-13487-6　　**OPEN**

# An effector protein of the wheat stripe rust fungus targets chloroplasts and suppresses chloroplast function

Qiang Xu[1], Chunlei Tang[1], Xiaodong Wang[1], Shutian Sun [1], Jinren Zhao[1], Zhensheng Kang[1]* & Xiaojie Wang [1]*

Chloroplasts are important for photosynthesis and for plant immunity against microbial pathogens. Here we identify a haustorium-specific protein (Pst_12806) from the wheat stripe rust fungus, *Puccinia striiformis* f. sp. *tritici* (*Pst*), that is translocated into chloroplasts and affects chloroplast function. Transient expression of *Pst_12806* inhibits BAX-induced cell death in tobacco plants and reduces *Pseudomonas*-induced hypersensitive response in wheat. It suppresses plant basal immunity by reducing callose deposition and the expression of defense-related genes. *Pst_12806* is upregulated during infection, and its knockdown (by host-induced gene silencing) reduces *Pst* growth and development, likely due to increased ROS accumulation. Pst_12806 interacts with the C-terminal Rieske domain of the wheat TaISP protein (a putative component of the cytochrome b6-f complex). Expression of *Pst_12806* in plants reduces electron transport rate, photosynthesis, and production of chloroplast-derived ROS. Silencing *TaISP* by virus-induced gene silencing in a susceptible wheat cultivar reduces fungal growth and uredinium development, suggesting an increase in resistance against *Pst* infection.

[1] State Key Laboratory of Crop Stress Biology for Arid Areas and College of Plant Protection, Northwest A&F University, Yangling 712100 Shaanxi, China. *email: kangzs@nwsuaf.edu.cn; wangxiaojie@nwsuaf.edu.cn

Plants are subjected to innumerable forms of stress, from extreme climate to microbial pathogens. Nevertheless, plant species survive and persist. This survival is due to multi-layer immune functions that diminish destructive attacks by potential pathogens and improve the defense systems of the plants[1]. The first layer of defense is the recognition of pathogen-associated molecular patterns (PAMPs), such as flagellin Flg22 and chitosan, by multiple pattern recognition receptors (PRRs) at the cell surface. This pattern recognition is usually associated with callose depositions, production of reactive oxygen species (ROS), activation of the miRNA pathway, a series of downstream signal transduction events, and expression of defense-related genes, all of which are termed PAMP-triggered immunity (PTI)[1–3]. Although PTI is prevalent in plants, preventing pathogenic invasion, these pathogens also possess the ability to secrete effectors to suppress the basal immune response[1]. As a typical example, the bacterial effector AvrPto physically interacts with plant FLS2, a receptor kinase recognized by Flg22, inhibiting the kinase activity of PRRs and suppressing host PTI[4]. In addition, plants have a group of specific disease resistance genes (R genes) in this defense response layer. R genes usually encode intracellular nucleotide-binding leucine-rich repeat (NB-LRR) proteins to detect and recognize pathogen-secreted proteins (effectors), leading to effector-triggered immunity (ETI)[5]. The strong recognition between effectors and R proteins usually results in the hypersensitive response (HR), a type of programmed cell death (PCD) that occurs at infection sites to prevent further expansion of pathogens.

Plant chloroplasts are vital photosynthetic organelles that supply energy for the synthesis of sugar and generate defense signals for plant immunity, such as ROS, salicylic acid (SA), and nitrous oxide (NO)[6,7]. SA biosynthesis occurs mainly in chloroplasts, and SA acts as a key signaling molecule by mediating other hormone signaling pathways and systemic acquired resistance[8,9]. In addition, when organisms encounter biotic or abiotic stress, the chloroplasts serve as major sensors to communicate with other organelles, including the mitochondria, nucleus, and peroxisome[10]. The photosynthetic electron transport chain consists of photosystem II (PSII), photosystem I (PSI) and cytochrome (Cyt) b6/f components. Light energy is captured by the reaction center chlorophyll (Chl) P680 and transferred to PSI, which is accompanied by the production of ATP and NADPH, which are important for the functions of other organelles, metabolic reactions, and growth cycles. However, ROS, as byproducts in chloroplasts, are also critical and effective components for plant immunity. ROS not only block further colonization by pathogens but also act as signaling molecules to reprogram nuclear defense gene expression[11]. To weaken the function of chloroplasts, some pathogens directly target host chloroplasts, altering the thylakoid membrane structure and suppressing the production of defense signals, including SA, NO, and ROS[12]. Several studies have provided insight into chloroplast-targeting effectors from bacterial pathogens. In Pseudomonas syringae, HopI1 can form a complex with the plant protein Hsp70 and recruit cytosolic Hsp70 to chloroplasts to inhibit Hsp70 activity[13,14]. The bacterial effectors HopK1 and AvrRps4 also target chloroplasts, and disruption of the localization of these effectors to the chloroplast led to loss of virulence[15]. In biotrophic fungi, Mlp124111 and Mlp72983 from Melampsora larici-populina were identified as chloroplast-targeted effectors when expressed in A. thaliana[16,17]. However, their exact functions are not clear. In comparison with bacterial effectors, our knowledge regarding the function of chloroplast-targeting effectors in biotrophic pathogens is limited.

Similar to many other obligate pathogens, Puccinia striiformis f. sp. tritici (Pst), an important pathogen in all wheat-growing areas[18], forms specialized infectious structures known as haustoria that are regarded as a bridge between the rust fungus and host. In addition to absorbing essential nutrients from the host[19], haustoria are known to produce and release various effectors to suppress defense responses in obligate fungi[20,21]. However, due to the difficulties in transient expression in hexaploid wheat and the lack of a stable transformation system for Pst, functional analyses of the Pst effectors have been severely hindered. Recently, several approaches have been developed to characterize the functions of genes in rust fungi, including heterologous expression systems, host-induced gene silencing (HIGS), and the effector-to-host analyzer (Ethan)[22,23]. Pseudomonas strains have been important model systems for identification of the function of biotrophic pathogen effectors[23]. All of the above techniques promote the development of the effector biology of biotrophic pathogens and provide the opportunity to further study the functions of Pst effectors, including Pst_8713. Pst_8713 can suppress plant PTI and ETI responses and contributes to the enhancement of Pst virulence using a heterogenous transient expression system[24]. However, in comparison with other pathogens, the molecular mechanism of how Pst effectors interfere plant defense response remains under-investigated. Recently, the PgtSR1 effector from Puccinia graminis f. sp. tritici (Pgt) that is closely related to Pst was shown to function as a fungal RNA-silencing suppressor, altering the abundance of small RNAs to regulate plant basal defenses and the ETI response to contribute to the virulence of pathogens[25].

To better understand the roles of effectors in Pst pathogenesis, in this study, we functionally characterized one putative effector, Pst_12806, which has a predicted chloroplast transit peptide[26]. Pst_12806 was highly expressed during Pst infection and it could suppress basal immunity in plants. Silencing of Pst_12806 by HIGS reduces fungal growth and disease development. Pst_12806 interacts with TaISP, a subunit of Cyt b6/f that connects PSII and PSI in the photosystem. The binding of Pst_12806 to TaISP impaires photosynthesis and reduced ROS accumulation, which may affect the function of the Cyt b6/f complex in the electron transport chain in vitro. Overall, our results show that Pst_12806 is translocated into chloroplasts and perturbs photosynthesis, avoiding triggering cell death and supporting pathogen survival on live plants, indicating the importance of interfering chloroplast functions in a biotrophic pathogen like Pst.

## Results

### Pst_12806 encodes a chloroplast-targeting protein. To better understand the virulence and molecular mechanisms of the rust fungus–wheat interaction, we sequenced the Pst isolate CYR32 and analyzed the secretome of this isolate[26]. Pst_12806 is a highly expressed gene that encodes small, secreted proteins with characteristics of fungal effectors. In comparison with the level in urediniospores, the expression level of Pst_12806 was upregulated over 50-fold at 24 h post inoculation (hpi), the crucial stage of haustorium formation and suppression of plant defense, based on qRT-PCR analysis (Supplementary Fig. 1). Then, the expression level of this gene decreased but remained higher at 48 hpi compared with that in urediniospores. To verify the function of the signal peptide of Pst_12806, Pst_12806SP (the signal peptide of Pst_12806), Pst_12806^ΔSP, and Avr1bSP (the signal peptide of Avr1b) were introduced into pSuc2t7M13ori and transformed into yeast strain YTK12 lacking a secreted invertase. Transformants were streaked on CMD-W plates and YPRAA plates, which only support the growth of yeast with secreting invertase[27]. Like the positive Avr1bSP, the transformant containing Pst_12806SP grew on the YPRAA plates, but the transformant containing Pst_12806^ΔSP and the empty vector failed to grow on the YPRAA plates (Supplementary Fig. 2a). The enzyme activity

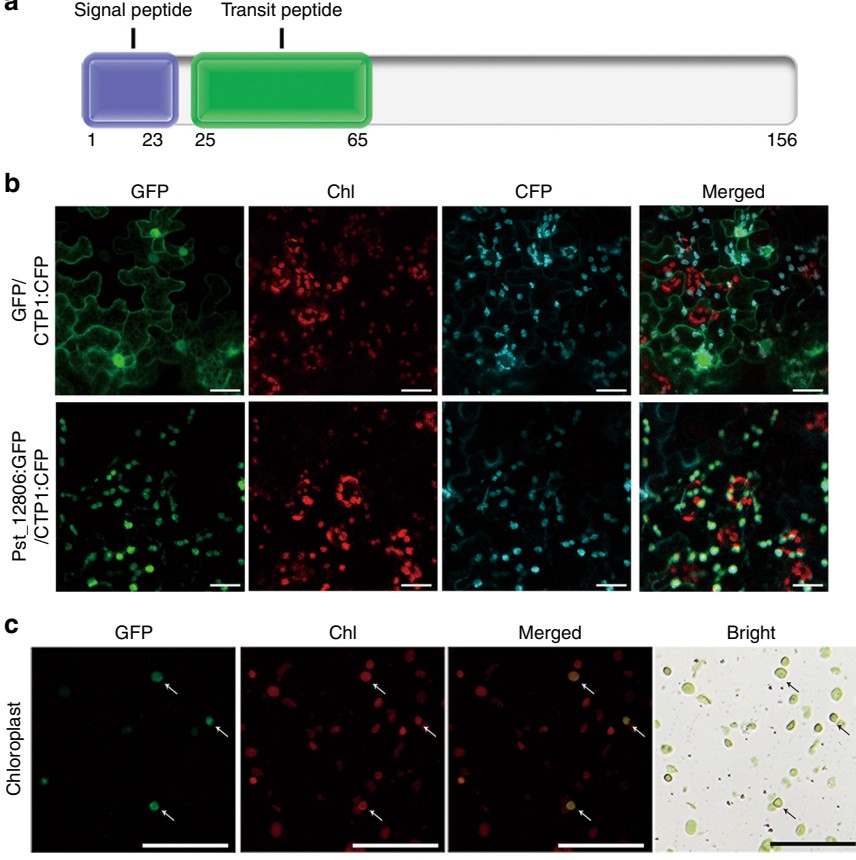

**Fig. 1 Pst_12806 accumulates in plant chloroplasts. a** Pst_12806 protein is predicted to have a signal peptide (1–23 aa) and a transit peptide (25–65 aa) by the SignalP 4.1 and LOCALIZER program. **b** Leaf tissues of *N. benthamiana* transiently co-expressing the Pst_12806$^{\Delta SP}$:GFP and CTP1:CFP or GFP alone and CTP1:CFP were examined by epifluorescence microscopy. Chl, chlorophyll. Bar = 50 μm. **c** Chloroplasts isolated from *N. benthamiana* leaves transiently expressing Pst_12806$^{\Delta SP}$:GFP fusion were examined by epifluorescence and bright field (BR) microscopy. The supernatant containing cytoplasm residues separated from chloroplast precipitates were examined as the control. Arrows point to chloroplasts. Bar = 50 μm.

of secreted invertase was also detected by the reduction of 2, 3, 5-triphenyltetrazolium chloride (TTC) and the secreted invertase of the transformant containing Avr1bSP and Pst_12806SP were detected by TTC assay, but the transformant containing *Pst_12806$^{\Delta SP}$* and the empty vector not (Supplementary Fig. 2b). These results support the functionality of the signal peptide of Pst_12806.

To verify the predicted chloroplast-targeting sequence (cTP; 25–65 aa) in the N-terminal region (Fig. 1a), the Pst_12806$^{\Delta SP}$: green fluorescent protein (GFP) (lacking the signal peptide) construct driven by the CaMV35s promoter was generated and transiently expressed in *N. benthamiana*. Mlp107772 (CTP1), which was reported to localize into chloroplasts[28,29], was synthesized and used as a marker protein to localize into chloroplasts (Fig. 1b). In the resulting tobacco cells expressing Pst_12806$^{\Delta SP}$: GFP, GFP signals were observed in chloroplasts (Fig. 1b). In tobacco cells expressing GFP alone as the control, GFP signals were observed in the cytoplasm and nucleus (Fig. 1b). GFP fusion proteins were observed in chloroplasts isolated from *N. benthamiana* leaves expressing Pst_12806$^{\Delta SP}$: GFP (Fig. 1c). These data showed that Pst_12806 encodes a chloroplast-targeting protein.

To determine the role of the predicted cTP sequence, we generated the cTP: GFP construct by fusing residues 23–65 of Pst_12806 with GFP. Transient expression of cTP: GFP resulted in tobacco cells with weak GFP signals in chloroplasts (Supplementary Fig. 3a). As a control, we also generated the Pst_12806$^{65–156}$: GFP construct and GFP: Pst_12806$^{\Delta SP}$. In

tobacco cells transiently expressing Pst_12806$^{65–156}$: GFP and GFP: Pst_12806$^{\Delta SP}$, GFP signals accumulated mainly in the cytoplasm and nucleus instead of chloroplasts (Supplementary Fig. 3a), indicating that the chloroplast transit peptide of Pst_12806 is required for the localization of this protein to the chloroplasts.

To determine whether the cTP peptide was cleaved when Pst_12806 was directed into chloroplasts, we generated the GFP: Pst_12806$^{\Delta SP}$ and Pst_12806$^{\Delta SP}$: GFP constructs and transiently expressed these constructs in *N. benthamiana*. The anti-GFP antibody detected a band of ~37 kD in tobacco cells transiently expressing Pst_12806$^{\Delta SP}$: GFP, which is similar to the predicted size of Pst_12806$^{65–156}$: GFP (Supplementary Fig. 3b). In contrast, a 43-kD band was detected in tobacco cells expressing GFP: Pst_12806$^{\Delta SP}$ (Supplementary Fig. 3b), indicated that GFP at the N-terminus of transit peptide might cover the transit peptide of Pst_12806. These results suggested that Pst_12806 protein is likely cleaved at the chloroplast transit peptide after being transported into chloroplasts.

**Pst_12806 suppresses cell death in plant.** Because *Pst* is not amenable to transformation, we assayed the ability of Pst_12806 to suppress Bax-induced PCD, which resembles the defense-related HR in plant cells[30,31]. Suppression of HR is usually regarded as one of the criteria for suppression of plant immunity[32]. Similar to Avr1b, an oomycete effector used as the positive control[24], Pst_12806 and Pst_12806$^{\Delta SP}$ suppressed Bax-induced PCD in *N. benthamiana* (Fig. 2a). As the negative control, eGFP

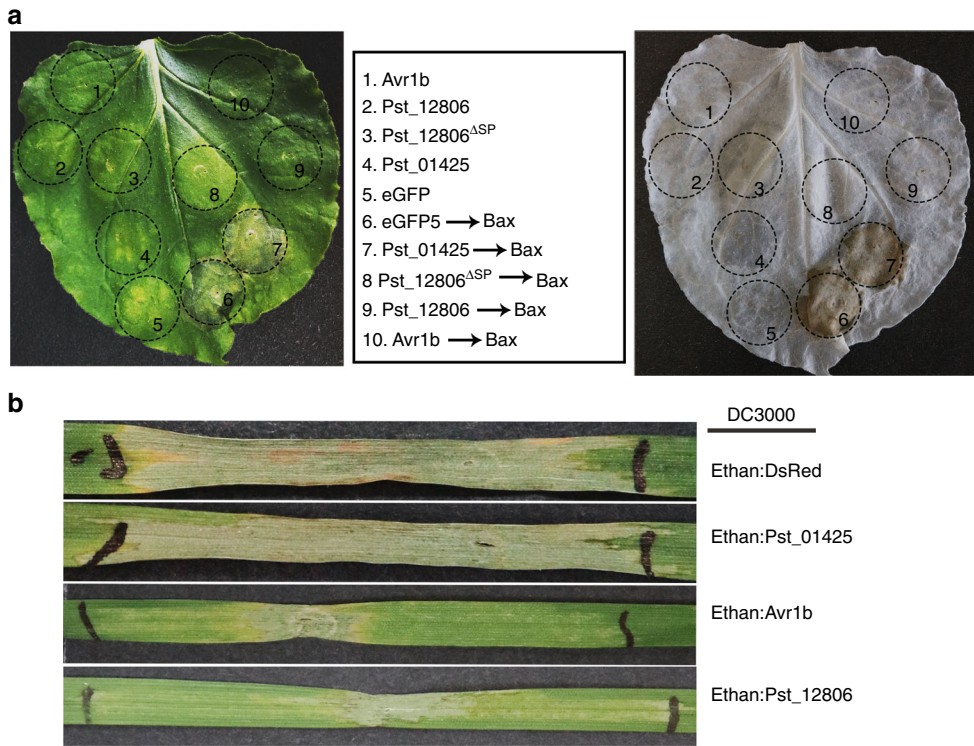

**Fig. 2 Suppression of Bax- and bacteria-triggered cell death by Pst_12806. a** Suppression of Bax-triggered cell death on *N. benthamiana* leaves. Pst_12806 was transiently expressed in leaves 24 h prior to infiltration with cells expressing Bax. The same leaf was examined before (left) and after (right) being decolorized with the destaining solution. At spots 1–10, agrobacterium cells expressing the labeled proteins (middle) were used for infiltration assays. **b** Transient expression of Pst_12806 delayed HR induced by bacteria in wheat. The second leaves of cultivar Chinese Spring seedlings were infiltrated with labeled *P. fluorescens* strains ($OD_{600} = 1.0$) and then inoculated with *P. syringae* strain DC3000 ($OD_{600} = 0.4$). Representative images were photographed 3 days after inoculation with DC3000. The boundaries marked with black lines were the regions inoculated with bacteria.

and Pst_01425 did not abolish the function of Bax (Fig. 2a). After removal of chlorophyll from tobacco leaves, macroscopic cell death was observed in the negative zone (Fig. 2a). Western blot analysis with anti-HA and anti-Bax antibodies confirmed the expression of these proteins in vivo (Supplementary Fig. 4).

To use the bacterial type III secretion for effector delivery[33], we expressed Pst_12806 and DsRed in the defective *Pseudomonas fluorescens* Ethan strain that lacks most of the effector clusters in the genome. As shown in Supplementary Fig. 5, infiltration of Ethan carrying Pst_12806 and DsRed did not elicit any phenotypic response in tobacco leaves, while significant cell death was observed after *P. syringae* DC3000 infiltration. Inoculation with Ethan carrying Pst_12806, but not DsRed, suppressed nonhost HR cell death induced by *P. syringae* DC3000 at $OD_{600}$ values of 0.1 and 0.2 (Supplementary Fig. 5). We then used *P. fluorescens* Ethan to deliver Pst_12806 into wheat. As expected, DC3000 elicited a strong HR on wheat leaves (Fig. 2b). The positive control Ethan: Avr1b suppressed the HR on wheat leaves, but Ethan:DsRed and Ethan: Pst_01425 did not have the same effect. Pst_12806 delivered from Ethan also decreased the nonspecific HR on wheat leaves co-infiltrated with DC3000 (Fig. 2b). Taken together, these results show that Pst_12806 has the ability to suppress plant cell death induced by Bax and bacteria, possibly by interfering with plant immunity.

**Pst_12806 suppresses the basal immune response in plant.** Pathogen effectors tend to suppress cellular defense associated with PTI, including the deposition of callose triggered by conserved PAMPs such as Flg22[34]. For PTI suppression assays with *N. benthamiana*, the Pst_12806$^{\Delta SP}$: GFP construct, GFP, and cTP: GFP were transiently expressed in leaves by agroinfiltration

followed by injection with 20 μM Flg22 at 48 hpi. At 12 h post infiltration with Flg22, aniline blue staining showed less callose accumulation in *N. benthamiana* leaves expressing Pst_12806$^{\Delta SP}$: GFP than in leaves expressing GFP alone and cTP:GFP (Fig. 3a). In repeated experiments, Pst_12806$^{\Delta SP}$: GFP significantly suppressed callose foci, but cTP: GFP could not (Fig. 3b). After Flg22 infiltration, the expression levels of the pathogenesis-related genes *NbPR1a*, *NbPR2*, and *NbWRKY12*[35,36] were threefold, threefold, and twofold lower, respectively, in tobacco leaves expressing Pst_12806$^{\Delta SP}$:GFP than in the control plants expressing GFP alone (Fig. 3c). But in leaves expressing cTP:GFP, the expression levels of these genes had no effect compared with the control plants (Fig. 3c). These results indicate that Pst_12806 suppresses plant defense responses.

To directly measure the ability of this protein to suppress plant immunity, we transiently expressed Pst_12806$^{\Delta SP}$ in wheat using the *P. fluorescens* Ethan-mediated delivery system and measured callose deposition caused by the nonpathogenic bacterium strain. Abundant callose deposits were observed in wheat leaves challenged by Ethan carrying the bacterial avirulence gene *AvrRpt2* and DsRed (Supplementary Fig. 6a), indicating the triggering of PTI in wheat cells. In repeated assays, transient expression of Pst_12806$^{\Delta SP}$ significantly reduced the amount of callose deposition compared with that in the control plants (Supplementary Fig. 6b). Taken together, these results suggest that Pst_12806 suppresses callose deposition and the basal immune response in wheat plants.

**Pst_12806 is required for the full virulence of *Pst* in wheat.** To further determine the role of *Pst_12806* during the wheat–*Pst* interaction, we knocked down *Pst_12806* expression by the barley

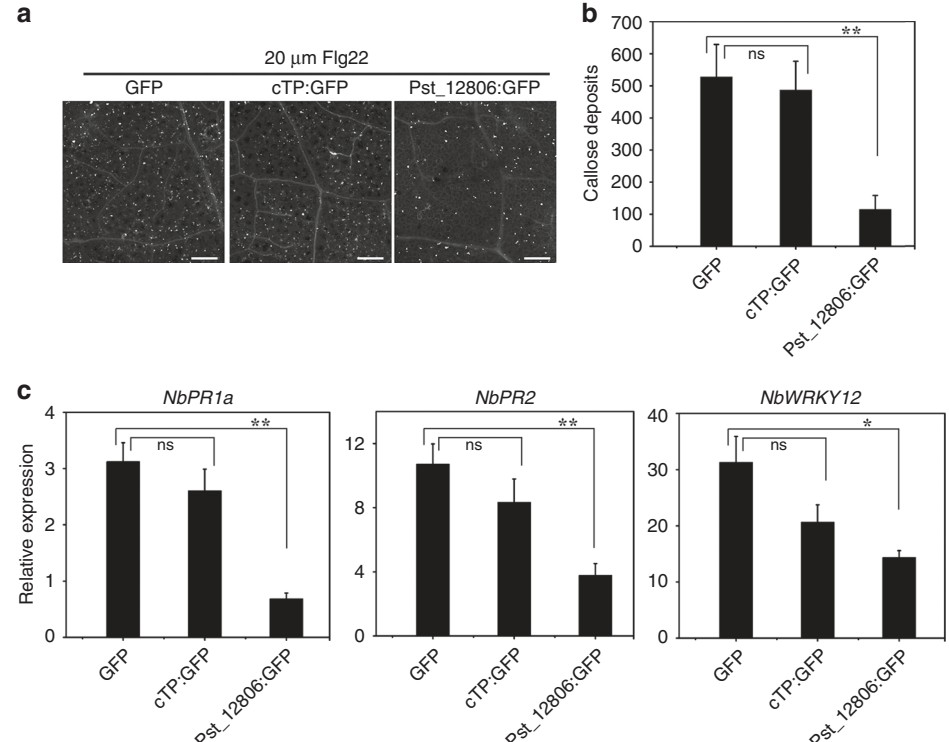

**Fig. 3 Suppression of PAMP -triggered PTI. a** Callose deposition induced by 20 µM Flg22 in *N. benthamiana* transiently expressing Pst_12806$^{\Delta SP}$:GFP,cTP: GFP or GFP clone. The representative images were captured at 12 h after infiltration with Flg22. Bar = 200 µm. **b** Pst_12806 suppresses callose deposits in *N. benthamiana*. The number of callose spots per 1-mm$^2$ areas was analyzed with the ImageJ software. Mean and standard deviation (SD) were calculated with results from three biological replicates. Asterisk (*) marks indicate significant difference (*$P < 0.05$; **$P < 0.01$) based on unpaired two-tailed Student's *t* test. **c** The relative expression levels of *PR1, PR2,* and *WRKY12* in *N. benthamiana* leaves transiently expressing Pst_12806$^{\Delta SP}$:GFP,cTP:GFP or GFP clone after infiltration with Flg22 were assayed by qRT-PCR with the *NbAct1* as a reference gene for normalization. Mean and standard deviations were calculated from three biological replicates. An asterisk indicates significant differences based on unpaired two-tailed Student's *t* test with the *P* values marked (*$P < 0.05$; **$P < 0.01$, ns not significant).

stripe mosaic virus (BSMV)-mediated HIGS assay. At 10 days after BSMV inoculation, while mild chlorotic mosaic symptoms were observed on BSMV-infected leaves, photobleaching was observed on plants inoculated with BSMV:TaPDS (Fig. 4a), indicating the efficacy of virus-induced gene silencing (VIGS) with BSMV. On wheat plants expressing the HIGS construct of *Pst_12806*, pustules continued to form on leaves infected by *Pst* at 14 days post infection (dpi) (Fig. 4b). However, the number of uredinia per 1.5 cm$^2$ decreased on plants expressing the HIGS construct compared with the control plants (Fig. 4c). In comparison with the control plants, the *Pst* biomass was significantly reduced in *Pst_12806*-silenced plants (Fig. 4d). Moreover, the hyphal length and infection areas of *Pst* at 24 hpi and 48 hpi also decreased by silencing of *Pst_12806* by HIGS (Fig. 4e, Supplementary Fig. 7a). In comparison with the control plants, the expression level of *Pst_12806* decreased by 65–70% at 24 and 48 hpi in plants expressing the HIGS construct of *Pst_12806* (Supplementary Fig. 7b). These results indicated that the expression of *Pst_12806* was partially knocked down by HIGS, and silencing of *Pst_12806* reduced the virulence and restricted the growth and development of *Pst* on wheat plants.

**Pst_12806 interacts with the photosynthesis-related protein TaISP**. To further explore the mechanism by which Pst_12806 influences the plant immune response, we used Pst_12806 as the bait to screen the yeast two-hybrid (Y2H) library constructed with RNA isolated from *Pst*-infected wheat leaves. A total of 95 clones were identified as 40 plant proteins and 2 rust proteins (Supplementary Data 1). Due to the chloroplast sublocalization of

Pst_12806, we focused on chloroplast proteins. Among 40 interactants, there are six chloroplast-related interactants and all of them were tested to interact with Pst_12806. Finally, Only TaISP (accession: AAM88439) interacted with Pst_12806. The putative Pst_12806-interacting genes encode a 222-aa protein that is 76.5% identical to NbISP (accession: XP_016485041) from *N. benthamiana* (Supplementary Fig. 8a), which was closest to TaISP located on chromosomes 2B and 2D (the amino acid sequence of TaISP-2B is identical to that of TaISP-2D). NbISP is a photosynthesis-related protein that is involved in the electron transport chain in chloroplasts[37]. Similar to NbISP, TaISP has typical CytB6-F_Fe-S and Rieske domains (https://www.ebi.ac.uk/Tools/hmmer/).

The interaction between TaISP and Pst_12806 was first tested by Y2H assays with the GAL4 Y2H system. Due to the 99% similarity in the nucleotide sequences of three copies of *TaISP*, we cloned only one *TaISP* (chromosome 2B) from the cDNA of the wheat cultivar Suwon 11. Yeast cells expressing both pBD-Pst_12806 and pAD-TaISP grew on SD-Ade plates and exhibited beta-galactosidase activity (Fig. 5a). In Y2H assays, Pst_12806 also interacted with NbISP (Supplementary Fig. 8b). To further detect their physical association by bimolecular fluorescence complementation (BIFC) assays, the coding sequences of *TaISP* and *Pst_12806* were inserted into pSATN-nEYFP and pSATN-cEYFP, respectively, and transformed into *A. tumefaciens* C58-C1 strain, and then infiltrated into tobacco leaves. The fluorescence signals of the interaction between TaISP and Pst_12806 were observed after 48 hpi and it specifically observed in chloroplasts (Fig. 5b). In contrast, there were no fluorescence signals in

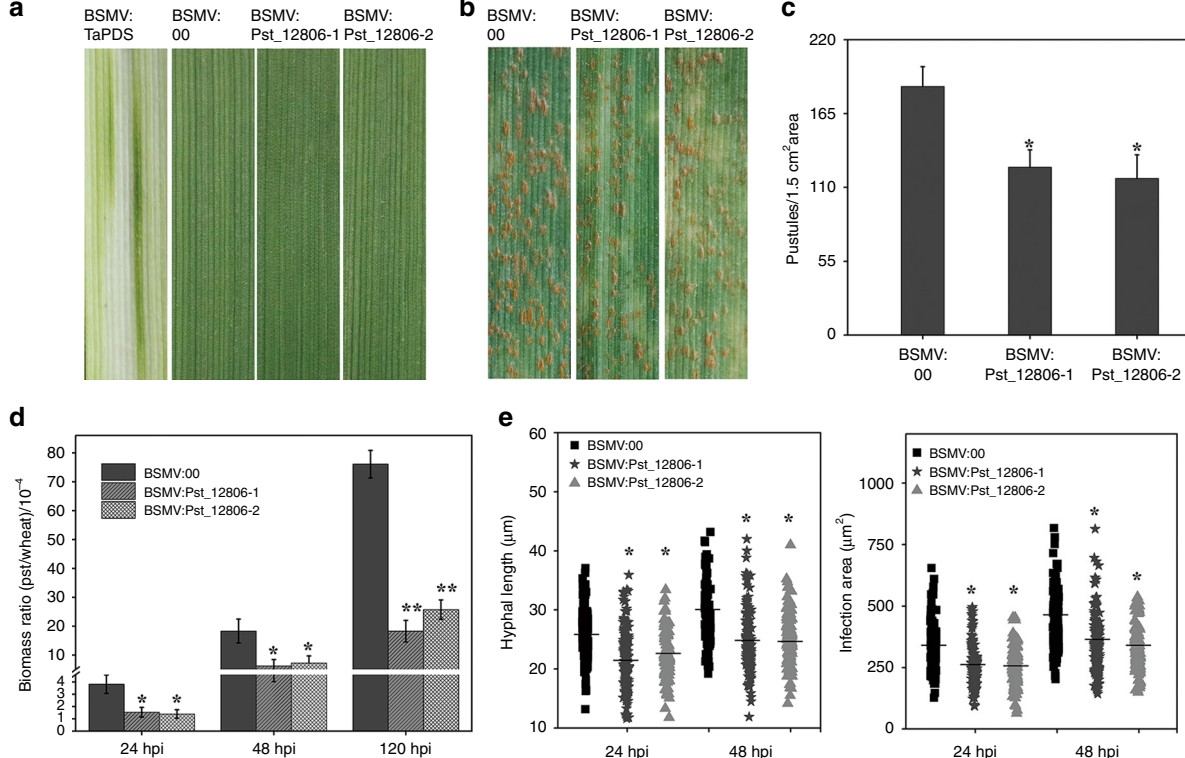

**Fig. 4 Silencing *Pst_12806* weakens the virulence of *Pst*. a** The second leaves of two-leaf stage wheat cultivar Suwon 11 were inoculated with barley stripe mosaic virus BSMV: *00* and recombinant BSMV:*TaPDS*, BSMV: *Pst_12806-1*, BSMV: *Pst_12806-2*, and the virus phenotypes were observed and photographed at 10 days after inoculation. **b** Seedlings of cultivar Suwon 11were inoculated with the labeled BSMV constructs on the second leaf for 10 days and then inoculated with urediniospores of *Pst* CYR31 on the fourth leaf. Leaves infected with *Pst* were examined at 14 dpi. **c** The number of uredinium pustules formed by *Pst* per 1.5 cm$^2$ on the fourth leaves of wheat plants treated with the labeled BSMV were scored at 14 dpi. Mean and standard deviation were calculated with results from three independent replicates, with 15–20 leaves examines in each replicate. The asterisks indicate significant difference ($P < 0.05$). **d** The *Pst*/wheat biomass ratio was assayed by qRT-PCR with RNA isolated from the fourth leaves of the same set of wheat plants as Fig.4c at 24, 48, and 120 hpi as described[66]. *TaEF-1α* and *PstEF-1α* were used to normalize the RNA level of wheat leaves and *Pst*, respectively. Mean and standard deviations were calculated with data from three independent replicates. The asterisks indicate significant difference in samples with *Pst_12806* silenced by HIGS in comparison with the control (*$P < 0.05$; **$P < 0.01$). **e** The hyphal length and infection area of *Pst* on the fourth leaves of the same wheat samples of 4 days were analyzed with CellSens Entry software at 24 and 48 hpi. Means were calculated from 50 infection sites of three biological replicates and were represented as solid lines in the picture. An asterisk indicates significant differences ($P < 0.05$) using unpaired two-tailed Student's *t* test.

chloroplasts when TaISP-nEYFP and cTP-cEYFP, and cTP-nEYFP and TaISP-nEYFP were transiently expressed, respectively (Fig. 5b). In addition, we confirmed the Pst_12806–TaISP interaction by co-IP assays in planta the TaISP:HA tag and Pst_12806:GFP fusion constructs were generated and transiently co-expressed in *N. benthamiana*. Total proteins were extracted from tobacco leaves and mixed with magnetic beads containing the GFP antibody. When detected with anti-HA and anti-GFP polyclonal antibodies, both TaISP and Pst_12806 were detected in western blots of total proteins and proteins eluted from GFP beads (Fig. 5c). To identify the Pst_12806-interacting region, we generated prey constructs of TaISP fragments containing the CytB6-F_Fe-S domain at the N-terminal or the Rieske domain at the C-terminal. Pst_12806 interacted with the Rieske domain but not the CytB6-F_Fe-S domain of TaISP in the Y2H assays (Fig. 5d). Therefore, these data confirmed that Pst_12806 physically interacted with TaISP in chloroplasts.

**Silencing of wheat *TaISP* enhances plant resistance to *Pst*.** To understand the function of *TaISP* during the wheat–*Pst* interaction, we first assayed the expression of *TaISP* at different *Pst* infection stages. Although the three copies of *TaISP* on chromosomes 2 AS, 2BS, and 2DS[38] share 99% similarity in nucleotide

sequence (Supplementary Fig. 9). We successfully designed PCR primers to specifically amplify individual TaISP alleles. When assayed by qRT-PCR, the *TaISP*-2B, and *TaISP*-2D transcripts were significantly induced in compatible interactions, approximately threefold at 24 hpi (Supplementary Fig. 10). In contrast, the expression level of *TaISP*-2A was not changed, suggesting that *TaISP*-2B and *TaISP*-2D played a role in the susceptibility of wheat to *Pst*.

To determine the role of *TaISP* genes in the resistance to *Pst*, we then silenced the transcription of all three copies of *TaISP* in the susceptible wheat cultivar Suwon 11 by BSMV-induced VIGS. When infected with the *Pst* isolate CYR31, fewer uredinia were produced on *TaISP* knockdown plants than on the BSMV control plants at 14 dpi (Fig. 6a). The transcription of the *TaISP* genes was reduced 60–70% at 24, 48, and 120 hpi in *TaISP* knockdown plants (Fig. 6b). The hyphal length and the infection areas were also reduced at 24 and 48 hpi in *TaISP* knockdown plants (Supplementary Fig. 11a, b). In addition, the *Pst* biomass was also significantly reduced at 120 hpi (Fig. 6c). Thus, we concluded that silencing of *TaISP* genes may limit *Pst* expansion in wheat leaves at late stages. In addition, we assayed the expression of three defense-related genes in *TaISP*-silenced plants. While *TaPR5* expression was not affected, the transcription of *TaPR1* and *TaPR2* was reduced 2–3-fold at 0 and 24 hpi in the *TaISP*

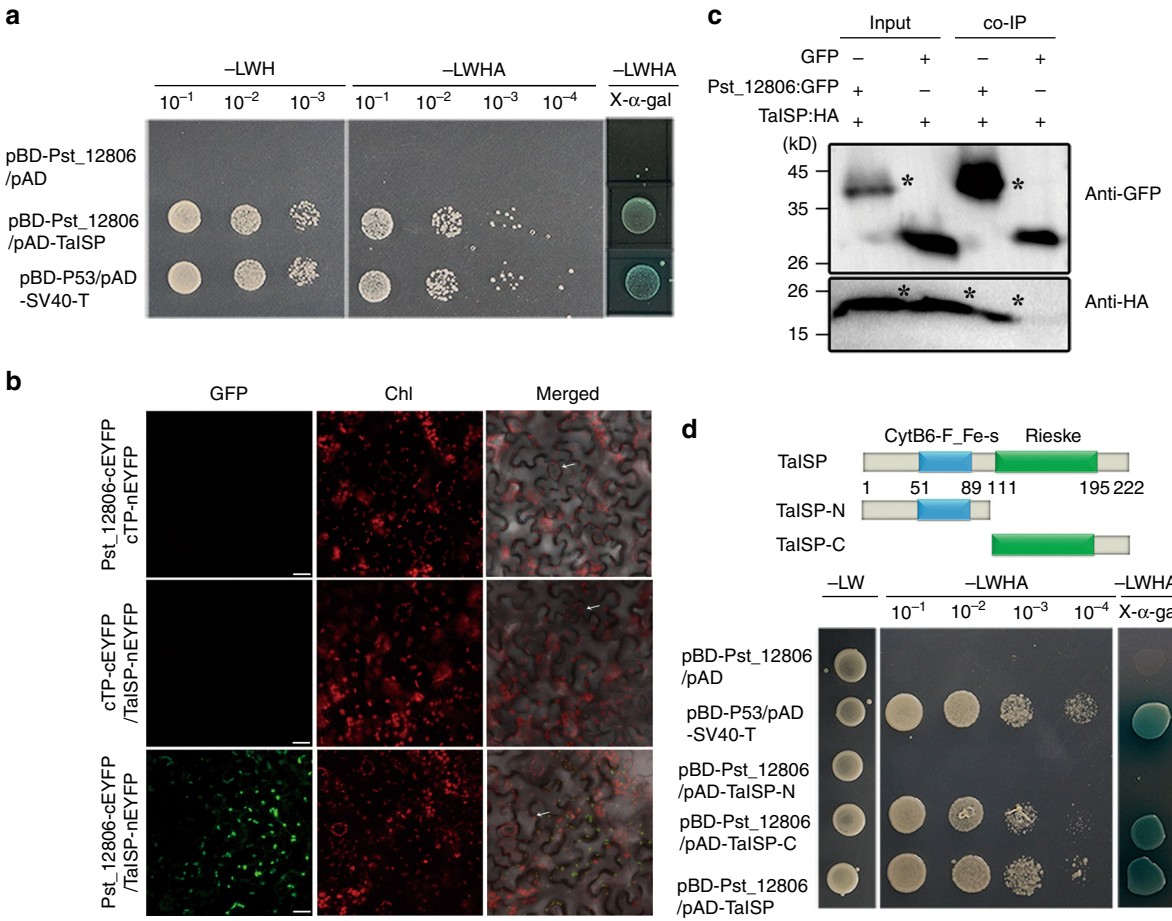

**Fig. 5 Pst_12806 interacts with TaISP in vivo. a** Detection of the interaction between Pst_12806 and TaISP by yeast two-hybrid assays. Yeast cells of AH109 strain transformed with the labeled constructs were assayed for growth on SD medium lacking LWH or LWHA and LacZ activities on SD-LWHA with X- α-gal. **b** Detection of the Pst_12806–TaISP interaction in chloroplasts of *N. benthamiana* leaves transiently expressing the marked constructs by bimolecular fluorescence complementation (BIFC) assays. These arrows marked guard cells. Bar = 20 μm. **c** Confirmation of the interaction between Pst_12806 and TaISP by co-immunoprecipitation assays. Western blots of total proteins from *N. benthamiana* leaves transiently expressing the marked constructs and proteins eluted from magnetic beads were detected with the anti-GFP or anti-HA antibody. The sizes of Pst_12806:GFP and TaISP:HA bands (marked with asterisk) were 41 and 24 kD, respectively. The protein marker was labeled on the left. **d** Yeast transformants expressing the labeled constructs of Pst_12806 and Rieske domains of TaISP were assayed for growth on SD-LWH or SD-LWHA and LacZ activities on SD-LWHA with X- α-gal. TaISP-N contains the N-terminal region of TaISP with the CytB6-F_Fe-S domain. TaISP-C contains the C-terminal region of TaISP with the Rieske domains. L leucine, W tryptophan, H histidine, A adenine, Chl chlorophyll.

knockdown plants (Fig. 6d), indicating that silencing of *TaISP* genes may affect the defense response in wheat.

**TaISP positively regulates photosynthesis in plant**. TaISP (also known as the Rieske Fe/S protein), a subunit of the Cyt b6/f complex, connects PSII and PSI of the photosynthetic electron transport chain. Because ISP is known to be a central member of the photosynthetic electron transport chains and reduced NbISP expression in transgenic tobacco significantly limited the electron transport rate (ETR) and $CO_2$ assimilation rate[39,40], we assayed the effects of transient expression of TaISP on the ETR in a heterologous tobacco system and in *TaISP*-silenced wheat plants. Compared with the control leaves expressing GFP, the ETR was increased at 6 and 72 h (Supplementary Fig. 12a), and non-photochemical quenching (NPQ) was reduced in tobacco leaves expressing *TaISP* at 6, 24, 36, and 72 h after infiltration (Supplementary Fig. 12b). In *TaISP*-silenced wheat plants, both the ETR and the magnitude of photochemical quenching (qL) decreased, although no considerable change in NPQ was observed (Fig. 6e). These results showed the influence of TaISP on the photosynthesis rate. The attenuated NPQ and increased qL

indicated reduced dissipation of excitation energy as heat caused either by energy-dependent quenching or photoinhibitory quenching[41].

**Pst_12806 attenuates photosynthesis and H₂O₂ accumulation.** Considering the localization of Pst_12806 to chloroplasts and the physical interaction of this protein with TaISP, we speculated that Pst_12806 may influence the electron transfer efficiency and $H_2O_2$ accumulation. To test this hypothesis, we measured the effect of Pst_12806 on photosynthesis by overexpressing this protein in *N. benthamiana* by infiltration. As shown in Fig. 7, overexpression of Pst_12806 led to a significant decrease in the ETR in tobacco leaves at 48 and 72 hpi compared with the ETR in the untreated control. As with negative control, no significant difference in ETR was observed in the leaves infiltrated with Ethan:DsRed (Fig. 7a). Moreover, the qL increased at 48 and 72 hpi, but NPQ increased at 36–72 hpi, in the Pst_12806-expressing plants (Fig. 7a). These results indicated that transient expression of Pst_12806 attenuates photosynthesis.

Because photosynthesis is always accompanied by $H_2O_2$ production, we measured ROS accumulation derived from

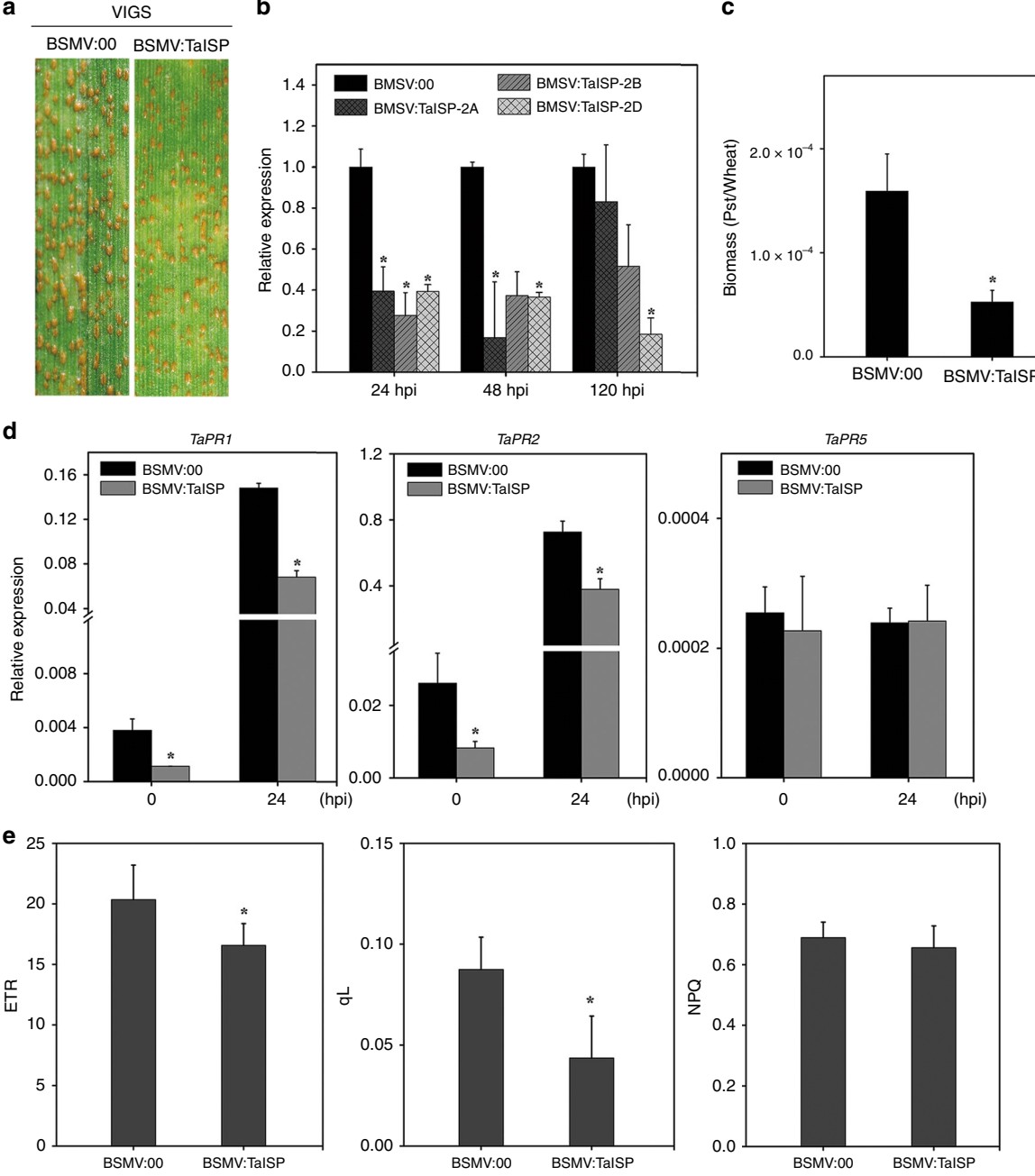

**Fig. 6 Knocking down *TaISP* increases resistance to *Pst* and weakens photosynthesis.** Seedlings of wheat cultivar Suwon 11 were inoculated with the empty BSMV:00 vector or recombinant BSMV:TaISP on the second leaves first for 10 days and then infected with *Pst* CYR31 on the fourth leaves. **a** *Pst* uredinia on the fourth leaves were examined at 14 dpi. **b** The relative expression level of *TaISP* in leaves inoculated with *Pst* were assayed by qRT-PCR at 24, 48, and 120 hpi. **c** The biomass ratio (*Pst*/wheat) was assayed with RNA isolated from the fourth leaves at 120 hpi. *TaEF-1α* and *PstEF-1α* were used to normalize the RNA level of wheat leaves and *Pst*, respectively. The experiments were repeated three times. An asterisk inidicates significant difference (*P* < 0.05, Unpaired two-tailed Student's *t* test). **d** Relative expression of the marked defense-related genes in the fourth leaves at 0 and 24 hpi. Transcript levels were quantified by qRT-PCR and normalized with *TaEF-1α*. The mean and standard deviation were calculated from three independent biological replications. The asterisk indicates the significant difference compared with GFP treated plants (*P* < 0.05, unpaired two-tailed Student's *t* test). **e** The fourth whea*t* leaves (*n* = 9) were collected at 10 days post inoculation with BSMV or BSMV:TaISP were assayed for photochemical quenching (qL), Non-Photochemical Quenching (NPQ), and electron transport rate (ETR). Mean values and standard deviation were assessed from three biological replications. An asterisk indicated the significant difference (*P* < 0.05, unpaired two-tailed Student's *t* test).

chloroplasts using the probe 2′7′-dichlorodihydrofluorescein diacetate ($H_2$DCF-DA) after bacterial infection of *N. benthamiana*[42]. Strong fluorescence was observed mainly in chloroplasts and cell membranes in the leaves infiltrated with Ethan:DsRed, confirming that the accumulation of ROS in response to bacterial infection is chloroplast derived (Fig. 7b). When we transiently expressed *Pst_12806* in tobacco, the oxidized dichlorofluorescein (DCF) signal accumulated in the cytoplasmic membrane instead of chloroplasts (Fig. 7b). To better observe ROS accumulation, we directly imaged samples from the lower epidermis. Compared with the control plants infiltrated with Ethan:DsRed, signals rarely appeared in chloroplasts on leaves infiltrated with Ethan:

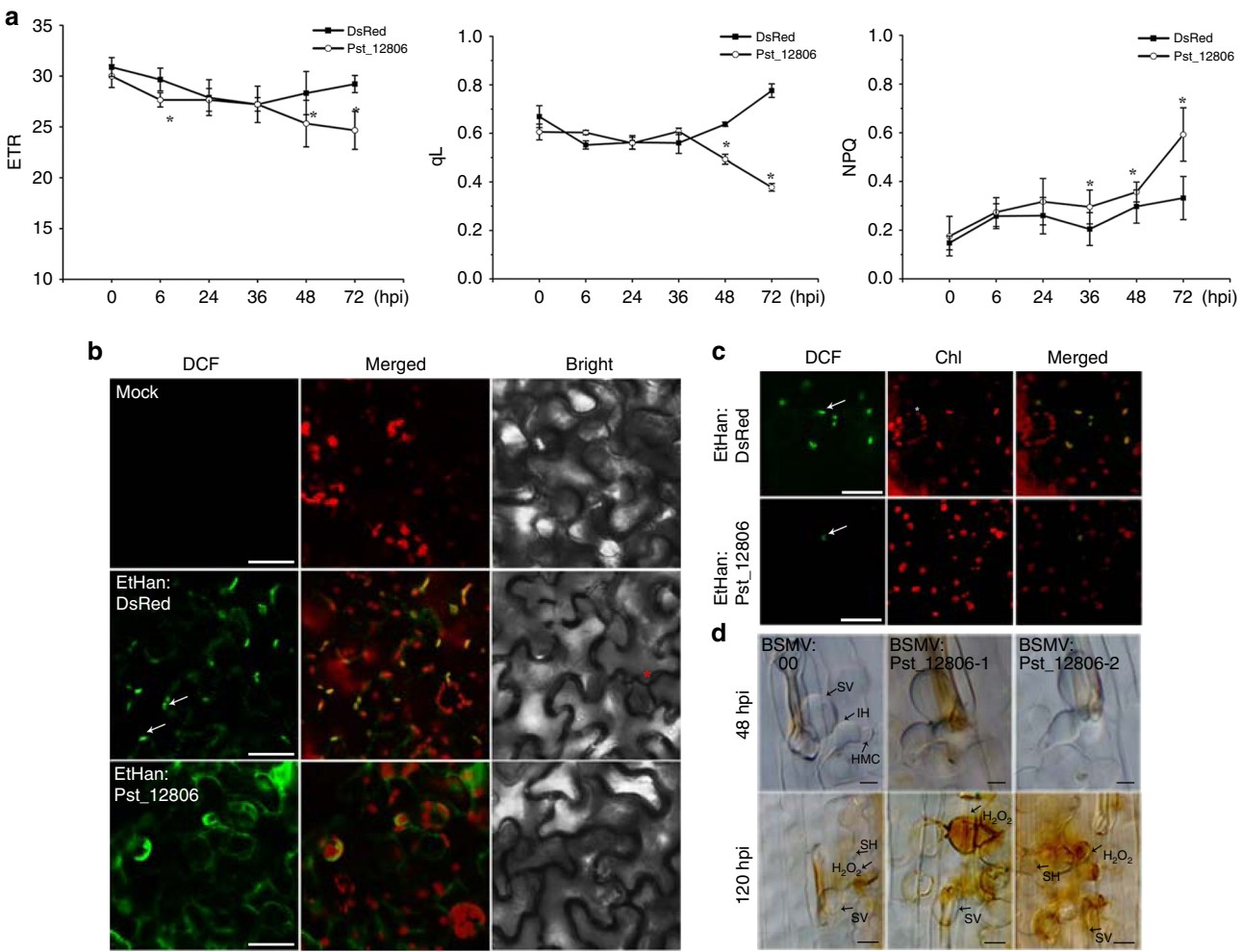

**Fig. 7 Pst_12806 suppresses photosynthesis and chloroplast-derived ROS. a** Assays for the values of electron transport rate (ETR), photochemical quenching (qL) and nonphotochemical quenching (NPQ) in *N. benthamiana* leaves infiltrated with Agrobacterium expressing Pst_12806 or DsRed (as the control). The mean and standard deviation of ETR, qL, and NPQ were calculated with data from six independent replicates. The asterisk indicates a significant difference ($P < 0.05$, unpaired two-tailed Student's *t* test). **b** *N. benthamiana* leaves treated with $H_2DCF$-DA were observed by fluorescence microscopy for DCF signals at 30–36 hpi after infiltration with bacteria expressing Pst_12806 or DsRed (control) or a mock solution. These arrows mark DCF signals in chloroplasts and asterisks mark guard cell, Bar = 50 μm. **c** ROS accumulation in chloroplasts was reduced in leaves transiently expressing Pst_12806. Tobacco leaves filtrated with *Agrobacterium* expressing Pst_12806 or DsRed treated with $H_2DCF$-DA were observed for DCF signals. Bar = 50 μm. Arrows indicate the DCF signals in chloroplasts. **d** Historical observation of $H_2O_2$ accumulation after DAB staining in wheat leaves of cultivar Suwon 11 with *Pst_12806* silencing by HIGS and inoculated with *Pst* strain CYR31 at 48 or 120 hpi. Bar = 10 μm. SV substomatal vesicle, IH infectious hyphae, SH secondary hyphae, HMC haustorial mother cell.

Pst_12806 (Fig. 7c). These results showed that Pst_12806 could hinder chloroplast-dependent ROS accumulation in response to nonpathogenic bacteria. In addition, we measured ROS production in Pst_12806-knockdown wheat seedlings of the cultivar Suwon 11 inoculated with the *Pst* CYR31. $H_2O_2$ accumulation was increased at 48 and 120 hpi in plants in which the expression of Pst_12806 was knocked down (Supplementary Fig. 7c), indicated that Pst_12806 can suppress the $H_2O_2$ burst in wheat mesophyll cells attached by *Pst*. Therefore, Pst_12806 may be involved in chloroplast metabolism for suppression of plant immunity, possibly by interfering with the generation of $H_2O_2$ in chloroplasts.

## Discussion

Chloroplasts function in primary metabolism and provide photosynthesis-derived carbon sources and energy. In recent years, increasing evidence supports a central role for chloroplasts in plant immunity via regulating the production of defense

molecules and chloroplasts may be targeted by secreted virulence factors of pathogens, such as HopN1 and HopI1 of *P. syringae*[43,44]. In this study, based on bioinformatic analysis of the secreted proteins of *Pst*, we found that Pst_12806 from the *Pst* isolate CYR32 possessed a chloroplast transit peptide and it was highly induced at early infection stages. The transit peptide of Pst_12806 was necessary and sufficient for targeting chloroplasts, and Pst_12806 entered the chloroplast and interacted with TaISP to suppress chloroplast-mediated immunity. The presence of these chloroplast-targeting effectors in the genome suggests that *Pst* and other *Puccinia* species could hijack the plant transport system to sort into chloroplasts and interfere with chloroplast functions as one of the strategies to suppress plant immunity. Several effectors targeting chloroplasts have been identified in *Puccinia* species by LOCALIZER prediction[45] or subcellular location. When expressed in *N. benthamiana*, two *Pst* effectors (Pst_03196 and Pst_18220) and two *Pgt* effectors (PGTG00164 and PGTG06076) were translocated into chloroplasts[45–47]. In

another rust fungus *Melampsora larici-populina*, CTP1 was shown to enter the stroma of chloroplasts with the help of its transit peptide[29]. Given the absence of chloroplasts in rust fungi, the chloroplast localization of these effectors strongly implies the potential roles in interfering with chloroplast functions.

Numerous studies in microbial pathogens have shown that effectors have evolved versatile patterns to target plant cellular processes and suppress plant immunity[48–51]. In this study, we showed that overexpression of *Pst_12806* in plants was suppressive to plant basal defense by disturbing the chloroplast function. In addition, hyphal growth and fungal colony areas were limited in *Pst_12806* knockdown wheat plants with increased ROS accumulation, resulting in a reduction in the number of *Pst* uredinia. Among other *Pst* effectors, PEC6 may inactivate TaADK, an adenosine kinase, to reduce SnRK1-related sugar levels and energy metabolism in response to biotrophic fungi[52]. Overexpression of *PSTha5a23* contributed to the virulence of *Pst*, but there was no change in the knockdown plant[53]. Silencing of *Pst_8713* could compromise the pathogenicity of *Pst* due to accumulation of host $H_2O_2$, although the molecular mechanism is unclear[24]. Therefore, like other microbial pathogens, *Pst* and other rust fungi may utilize the spatiotemporal deployment of a remarkably diverse range of effector proteins to control plant defenses and cellular processes.

ISP, a subunit of the Cyt b6/f complex, plays a positive role in photosynthesis and the production of ROS in chloroplasts and indirectly regulates the expression of plant defense genes[40]. Pst_12806 was shown to physically interact with a photosynthesis-related protein TaISP, which has not been functionally characterized in wheat. Our results showed that overexpression of *TaISP* promoted the ETR and had a strong positive effect on plant photosynthesis, and silencing of *TaISP* in wheat resulted in a reduction in the ETR and qL. Earlier studies on transgenic tobacco showed that a low level of ISP drastically limited the ETR and $CO_2$ assimilation rate[39,40]. In addition, chloroplasts are major generators of ROS as the byproducts of the electron transport chain. Our results showed that transient expression of Pst_12806 inhibited photosynthesis and the production of bacteria-induced ROS in chloroplasts. Excessive ROS accumulation is known to cause cell death at the infection site to hinder the uptake of nutrients required for growth and development of the pathogen[32]. Thus, the stripe rust fungus may secrete Pst_12806 to suppress plant cell death by reducing

chloroplast-derived ROS for their survival. To our knowledge, effectors targeting the ISP protein to disrupt the chloroplast function and suppress plant immunity have not been reported in microbial pathogens.

Domain deletion analysis showed that Pst_12806 interacted with the C-terminal region of TaISP containing the Rieske domain. The Rieske domain contains a [2Fe-2S] cluster coordinated by two cysteine residues and two histidine residues[54], which transfer an electron to the 2[Fe-2S] cluster, and then, the electron is released to the heme iron of cytochrome c or cytochrome f[37]. Therefore, we speculate that the binding with Pst_12806 may weaken the ability of the Rieske domain of TaISP to transfer electrons during photosynthesis and subverts photosynthesis and $H_2O_2$ accumulation in chloroplasts. Photosynthetic light reactions require the participation of four protein complexes in thylakoid membranes (PSII, the cytochrome b6f complex, PSI, and ATP synthase). It has been reported that bacterial effector HopN1 interacts with plant PsbQ, an element of the oxygen-evolving complex of PSII, to suppress chloroplast-derived ROS[43]. Our results showed that the Cyt b6/f complex was targeted by a *Pst* effector. Therefore, different photosynthesis components may be targeted by bacterial and fungal effectors to suppress photosynthesis and reduce photosynthesis-derived ROS and plant basal defense. Although it remains challenging to identify and characterize targets of candidate rust effectors, 33 additional secreted effectors in *Pst-130* are predicted to have the transit peptide and may also target different components of chloroplasts[45]. The genomes of the wheat and leaf rust fungi, two *Puccinia* pathogens (*Puccinia graminis* f. sp. *tritici* and *Puccinia triticina*) closely related to *Pst*, also have 60 and 42 candidate effectors with the predicted transit peptide, respectively[45]. Based on results from this study, it is tempting to speculate that *Pst* and other rust fungi or obligate pathogens may use various effectors targeting different components of chloroplasts to reduced photosynthesis and chloroplast-dependent plant immunity.

Biotrophic fungi feed on live plant cells and establish a close association with the host by development of haustoria for nutrient uptake[55]. Therefore, photosynthate is an important energy source for both the plant and pathogen. In *TaISP* knockdown wheat plants, the ETR and transcript levels of defense-related genes *PR1* and *PR2* were significantly reduced, which indicated that silencing of TaISP is inhibitory to photosynthesis and plant defense. However, the reduction in plant

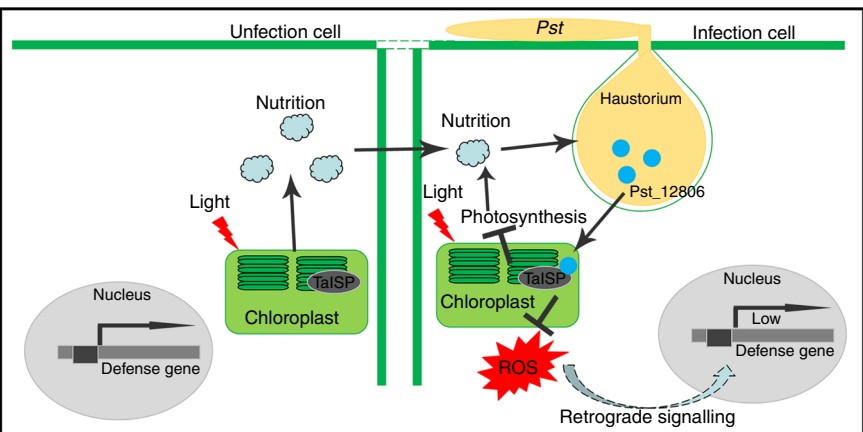

**Fig. 8 Working model of the function of Pst_12806 during *Pst*–wheat interactions.** In infected plant cells, the effector Pst_12806 is secreted from haustoria and translocated into chloroplasts, where it interacts with TaISP. Binding of Pst_12806 with TaISP interferes with the function of this protein in the electron transport chain and results in a decrease in the plant electron transport rate and attenuation of ROS accumulation, which in turn inhibits the expression of defense-related genes. In neighboring uninfected cells lacking Pst_12806, TaISP functions normally in photosynthesis, and synthesized carbohydrates may be transferred into the infected wheat cells.

photosynthesis is disadvantageous to nutrient uptake for obligate biotrophic fungi. Therefore, we inferred that impairment of photosynthesis by Pst_12806 reduces plant resistance at the expense of nutrient uptake, likely because of a low demand for carbohydrates at the early infection stage with limited infectious hyphae (Fig. 8). As expected, when the pathogen is faced with fate determination, survival is a priority, which the pathogen accomplishes by avoiding triggering cell death at infection sites at the early stage (Fig. 8). At late infection stages, pathogens establish a parasitic relationship with the host plant, and uptake of nutrients from host cells is highly important for sustainable growth and pathogen development. In several plant–pathogen interactions, along with decreased photosynthesis at the infection sites, enhanced photosynthesis is detectable in tissues surrounding the infected areas[56]. During the late stage of Pst infection, it is likely that the flow of carbohydrates is forcibly directed into infected wheat cells (sink cells) from neighboring uninfected cells (source cells), where TaISP functions normally in photosynthesis (Fig. 8). The stripe rust pathogen may keep the infected cells alive and primarily obtain nutrients from the adjacent uninfected cells for growth and development, which explains the typical green islands observed on the infected wheat leaves[6,57].

In conclusion, we have identified a Pst effector Pst_12806 that is translocated into host plant chloroplasts, where it interacts with the Rieske domain of TaISP and attenuates photosynthesis rate, decreases ROS accumulation at the infection sites and inhibits plant defenses. This study provides insights into the regulation of wheat photosynthesis by Pst, which will pave the way for understanding the molecular mechanism by which pathogen effectors manipulate chloroplast-targeted immunity and metabolism in chloroplasts.

## Methods

**Sequence analysis**. The Pst_12806 sequence was derived from the haustoria transcriptome of the virulent Pst pathotype CYR31. Genomic DNA was isolated from the urediniospores of CYR31 using the CTAB method[58], and total RNA was extracted from Pst-infected wheat leaves with the RNA Purification Kit (cat no. 74904, QIAGEN Bioinformatics). The DNA sequence of Pst_12806 was amplified from CYR31 genomic DNA (accession: GCA_000474995.1). The signal peptide of Pst_12806 was identified by using SignalP 4.0 (http://www.cbs.dtu.dk/services/SignalP/). The molecular size of Pst_12806 was predicted by the Compute pI/Mw tool (http://web.expasy.org/compute_pi/). The localization of the secreted protein was determined by LOCALIZER (http://localizer.csiro.au/index.html) and WoLF PSORT II (https://www.genscript.com/wolf-psort.html). Protein domains were analyzed using HMMER software (http://www.ebi.ac.uk/Tools/hmmer/). All the primers used in this study are described in Supplementary Data 2.

**Experimental materials and growth conditions**. N. benthamiana and wheat seedlings were grown in a greenhouse under 8/16 h night/day conditions at 22 °C and 16 °C, respectively. For bacterial material, Escherichia coli (DH5a) and Agrobacterium tumefaciens (GV3101) were stored at −80 °C and cultured on LB medium at 37 °C and 28 °C, respectively. The yeast strain AH109 was cultured at 30 °C for Y2H assays. P. fluorescens Ethan and P. syringae DC3000 were cultured on King's B medium at 28 °C. Suwon 11 was inoculated with urediniospores of Pst CYR31[59]. Fresh urediniospores were collected from infected plant leaves and stored at −80 °C for RNA or DNA extraction.

**Isolation of chloroplasts**. Three N. benthamiana leaves infected with A. tumefaciens GV3101 harboring Pst_12806:GFP were harvested 48 h after inoculation and cut into small pieces in a mortar. A total of 2 mL of cold separation solution was added, and the sample was ground to a paste. The suspension was filtered into a 15-mL tube through a double pad of gauze, and the liquid was centrifuged at $1000 \times g$ for 5 min at 4 °C. Then, 200 μL of the supernatant and the sediment was diluted with 200 μL of suspension buffer (0.35 M NaCl) and observed with an Olympus BX-51 fluorescence microscope (ocular: 10×; objective: 20×). The separation solution was composed of 330 mM sorbitol, 50 mM Tris-Cl (pH 7.6), 2 mM EDTA, 5 mM MgCl₂, 10 mM NaCl, and 2 mM sodium erythorbate.

**Transient expression of proteins**. For the Bax-induced cell death assay, genes were cloned into pGR106 and transformed into A. tumefaciens GV3101, which was cultured in Luria–Bertani medium with 50 mg L⁻¹ rifampicin and 50 mg L⁻¹ kanamycin. The method for infection of tobacco followed procedures[60]. Briefly, the

recombinant strains were washed three times with 10 mM MgCl₂ and infiltrated into leaves of 4-week-old N. benthamiana at an OD₆₀₀ of 0.4. The strains containing eGFP:HA, Avr1b:HA, Pst_01425:HA, Pst_12806ᐃSP:HA, or Pst_12806:HA were infiltrated into leaves 24 h prior to infiltrating the strain containing Bax. The infected leaves were harvested for protein extraction at 72 hpi. After 7 dpi, the leaves were decolorized in ethanol/acetic acid (1:1) for 5–7 days until they were translucent, and then, the leaves were photographed. For localization, GV3101 carrying the corresponding constructs were resuspended in acetosyringone (AS) buffer (10 mM MgCl₂, 10 μM AS, 10 mM 2-(N-morpholino) ethanesulfonic acid (MES), pH 5.6) at an OD₆₀₀ of 0.6 and injected into 4-week-old tobacco leaves. Then, GFP fluorescence was observed at 48 hpi by confocal microscopy.

For the wheat assay, Pst_12806 was introduced into pEDV6 and transformed into P. fluorescens and P. syringae, and the transformed bacteria were cultured in King's medium with appropriate antibiotics for 48 h and then collected and washed twice with 10 mM MgCl₂. P. fluorescens was resuspended at an OD₆₀₀ of 1.0, infiltrated into the second leaves, and then harvested at 24 hpi for observation and detection of callose. P. syringae was diluted to an OD₆₀₀ of 0.4, mixed with the DC3000 strain at a ratio of 1:1, and infiltrated into wheat leaves (Chinese Spring). Representative images of treated leaves were captured at 4 dpi. For the tobacco assay, Ethan containing Pst_12806ᐃSP or DsRed was infiltrated into leaves at an OD₆₀₀ of 1.0 prior to injection of DC3000 at an OD₆₀₀ of 0.2 or 0.1. The images were obtained at 5 dpi.

For the BIFC assay, the C-terminus of TaISP, cTP of Pst_12806, and Pst_12806 were ligated with YFP in the vectors pSATN-nEYFP and pSATN-cEYFP, respectively, and the constructs were transformed into A. tumefaciens C58-C1. The Agrobacterium strains were infiltrated at an OD600 of 0.5. Two days after inoculation, images were captured by confocal microscopy with a 488-nm laser.

For the co-IP assay, GV3101 carrying the Pst_12806:GFP (pCAMBIA1302), TaISP:HA (pICH86988), and P19 construct was co-infiltrated into N. benthamiana leaves. P19 protects exogenous genes from being silenced in tobacco leaves. The infiltrated leaves were harvested at 48 hpi and ground to a powder in liquid nitrogen and then homogenized in extraction buffer (10% glycerol, 25 mM Tris-HCl (pH 7.5), 1 mM EDTA, 150 mM NaCl, 2% polyvinylpolypyrrolidone (PVPP), 10 mM DTT, 1× protease inhibitor, and 1 mM PMSF). The extract was centrifuged at $15000 \times g$ for 15 min, and the supernatant was transferred into a fresh tube for the co-IP assay. The magnetic beads (cat. no. 10003D, Thermo Fisher Scientific) were washed three times with 500 μL of extraction buffer and incubated with rabbit anti-GFP (1:5000; cat. no. 50430-2-AP, Proteintech Group) at room temperature for 15 min according to the manufacturer's instructions. Then, the total protein solution was incubated with magnetic beads at 4 °C for 4 h, and the beads were collected by washing three times with 500 μL of Tris-HCl buffer and 0.5% Tween-20. Proteins bound to the magnetic beads were boiled for 10 min and detected by western blotting.

For immunodetection, the proteins on the PVDF membrane were detected by the corresponding mouse anti-GFP (1:5000; cat. no. sc-9996, Santa Cruz Biotechnology) and mouse anti-HA (1:5000; cat. no. H3663, Sigma-Aldrich) with a secondary goat anti-mouse IgG (whole molecule)-peroxidase-conjugate antibody (1:2000; cat. no. sc-516102, Santa Cruz Biotechnology).

**Transient expression of proteins in yeast strain**. The Matchmaker GAL4 system (CLONTECH Laboratories) was used to screen a cDNA library from pathogen-infected wheat following the Matchmaker Y2H system protocol (CLONTECH Laboratories). A binding domain fusion of Pst_12806 (pGBKT7-Pst_12806) (without signal peptide) and activation domain fusion of TaISP (pGADT7-TaISP) were co-transformed into the yeast strain AH109 and diluted on the corresponding medium (SD/-Leu/-Trp/-His and SD/-Leu/-Trp/-His/ -Ade) for selection of transformants.

For yeast secretion assay, the Pst_12806SP (the signal peptide of Pst_12806) and the truncated Pst_12806ᐃSP (without signal peptide of Pst_12806) were inserted into plasmid pSuc2t7M13ori and transformed into yeast strain YTK12[27]. Transformants were diluted and screened on CMD/-W medium and YPRAA medium plates. The invertase enzymatic activity was detected by the reduction of 2, 3, 5-triphenyltetrazolium chloride (TTC) to insoluble red colored 1, 3, 5-triphenylformazan (TPF). Transformants were cultured in liquid CMD/-W medium at OD₆₀₀ of 0.3 and about 1.5 mL of cell suspension was collected and resuspended in 250 μL of 10 mM acetic acid–sodium acetate buffer (pH 4.7), 500 μL 10 % sucrose solution (w/v) and 750 μL sterile distilled water at 37 °C for 10 min. Then 400 μL of the supernatant after centrifugation at $12,000 \times g$ for 1 min was put into glass test tube containing 3.6 mL 0.1% TTC solution at room temperature for 5 min. The picture was immediately obtained after colorimetric change.

**Histochemical assays**. For observation of callose deposits, Pst_12806 (pEDV6-Pst_12806) was infiltrated into wheat leaves using T3SS with an OD₆₀₀ of 0.4. The infected leaves of N. benthamiana and wheat were decolorized in destaining solution (absolute ethyl alcohol:acetic acid, 1:1 v/v) and then immersed overnight in chloral hydrate. Transparent leaf segments were stained with 0.05% aniline blue in 0.067 M K₂HPO₄ (pH 9.6)[23]. Callose deposits were analyzed in fields of 1 mm² using ImageJ software[61].

For observation of hyphae, the infected tissues were cleared with ethanol as described above and then autoclaved in 1.5 ml of 1 M KOH at 121 °C for 5–6 min. Those fragments were washed three times with 50 mM Tris-HCl (pH 7.4) for 15 min. The rinsed fragments were incubated in wheat germ agglutinin (WGA) Alexa-488 solution (cat. no. W11261, Thermo Fisher Scientific)[62]. Fifty infection sites were observed with an Olympus BX-51 microscope (ocular: 10×; objective: 20×) and CellSens Entry software (version: V1.7).

For the $H_2O_2$ detection assay, wheat leaves were incubated in DAB buffer under light for 8 h. After the DAB solution was absorbed, the samples were treated as above to remove chlorophyll until the samples became transparent. The materials were observed by bright-field microscopy, and the area containing $H_2O_2$ was calculated using CellSens Entry software. We also used 2′7′-dichlorodihydrofluorescein diacetate ($H_2$DCF-DA) as a probe to detect photosynthesis-derived ROS in tobacco. Pst_12806 was transiently expressed using the T3SS assay in tobacco after 24 hpi, and 10 μM $H_2$DCF-DA (Sigma-Aldrich, cat. no. D6883) was injected into leaves by using a syringe after 5–10 h[12]. Then, the samples were observed and photographed by microscopy (ocular: 10×; objective: 20×) under laser excitation at 488 nm and emission at 512–527 nm.

**Detection of chlorophyll fluorescence and ETR.** The chlorophyll fluorescence assay was performed using a PAM-2500 instrument (Heinz Walz Company, Germany) according to the manufacturer's instructions and a previously described protocol[63]. First, 4-week-old tobacco leaves were treated with Ethan: Pst_12806 and Ethan:DsRed, under 2000 LUX light. TaISP-GFP and GFP were transiently expressed in tobacco leaves and then dark-adapted for 20 min before chlorophyll fluorescence and ETR were measured at 19–20 °C. The ETR the TaISP knockdown plant was detected after 10 days of viral infection. Measurement of 8–10 leaves was performed per replicate, and three biological replicates were examined.

**BSMV-mediated gene silencing.** For silencing of Pst_12806 and TaISP, the recombinant vectors and the vector containing the tripartite viral genome were linearized and transcribed to RNA and then inoculated onto the second wheat leaves[64]. Transcripts of each vector (α, β, γ, or recombinant γ-gene) were mixed in a 1:1:1 ratio, and a total volume of 30 μL was added to 160 μL of FES buffer. Plants infected with virus were then grown at 25–27 °C. The plants inoculated with BSMV:TaPDS (phytoene desaturase) were used as an index for BSMV infection[64,65]. After 10 days, the fourth leaves were inoculated with Pst CYR31 on Suwon 11. Then, 15–20 infected leaves were assessed for phenotype identification and determination of urediniospore count. Fifty infection sites on the three leaves were analyzed for hyphal length and haustorium count. RNA was extracted from three leaves for assessment of silencing efficiency.

**qRT-PCR analysis.** Germ tubes were obtained from urediniospores germinated at 10–12 °C for 12–16 h in water. Pst_EF and TaEF were chosen as the internal control genes. All reaction products of qRT-PCR were analyzed using BIO-RAD CFX software according to the manufacturer's directions and qRT-PCR application guides. Biomass was determined by absolute quantification using double-standard curves[66]. The wheat cDNA and Pst cDNA were diluted in a gradient series at 10×, 20×, 50×, 100×, 200×, 500× and 1000×. The coefficients for the double-standard curves were >0.99, and the slope of these curves as approximately −3.3. The biomass of wheat and Pst was indicated by the internal control genes Ta_EF and Pst_EF, respectively.

**Reporting summary.** Further information on research design is available in the Nature Research Reporting Summary linked to this article.

## Data availability
GenBank accession codes include AAM88439 for TaISP genes, XP_016485041 for NbISP gene, KNE93802 for Pst_12806. Data supporting the findings of this work are available within the paper and its Supplementary Information files, or from the corresponding author upon request.

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

## Acknowledgements

We thank Prof. Jinrong Xu (Purdue University) for critical reading and revision of this article. We also thank Duplessis Sébastien from INRA Centre Grand-Est for useful discussions. This study was supported by the National Natural Science Foundation of China (31430069, 31972352, 31422043, and 31772116), the National Key Research and Development Program of China (2018YFD0200408), Shaanxi Innovation Team Project (2018TD-004), the Central Human Resource Department "Ten-thousand Program", and China Agriculture Research System (CARS-3).

## Author contributions

X.J.W., Q.X., and C.T. designed the research. Q.X. conducted most of the experiments. J.Z. and S.S. constructed vectors and cultured tobacco and wheat plants. X.D.W. performed the photosynthesis assay and BIFC assay. Q.X., C.T., and X.J.W. analyzed the data, Q.X., C.T., Z.K., and X.J.W. wrote the article.

## Competing interests

The authors declare no competing interests.
