## [Peer Review File · Nature Communications]

Reviewers' comments:

Reviewer #1 (Remarks to the Author):

In their manuscript the authors present an effector protein from the wheat stripe rust that targets chloroplast mediated host immunity. The authors provide ample evidences that the effector target the chloroplast, the experiment all have appropriate control, most of the conclusion are sound and the manuscript is very well presented.

All together I have only two major issue which I would like better explained in the manuscript, other issues are minor.

Major issue 1). In line 227 the authors mention that 95 Y2H clones were recovered, from which two partial TaISP sequences were recovered. Hence the rest of the manuscript focuses on the TaISP. It seems that it needs to be justified why the TaISP was selected and not any other interacting protein from the list of 40 interactants, if not it really looks as cherry picking. In addition (I do not blame the authors for that), but the supplementary data is very hard to analyse, I think the submission portal has separated the columns of Y2H results which appear on separate pages, this would have to be solved prior to publication.

Major issue 2: the title states ...suppress chloroplast-mediated host immunity.

I agree that the data presented in the manuscript clearly demonstrate that chloroplast function and H₂O₂ accumulation are affected. However, these processes are not strictly and only related to host immunity, for this reason I believe the title goes to far. Saying it suppresses chloroplast function would be OK, but what is the evidence that immunity specific aspects of the chloroplast are suppressed ?

Minors

Line 87. In ref 16 expression was in *A. thaliana* not *N benthamiana*

Line 133. compared to that (not to than)

Line 179. Indicate EthanDsRed in the figure

Line 297. As with the negative...

Line 309 to 318. There are discrepancies between the letters indicated in the text and the labeling of the panels of the figure.

Reviewer #2 (Remarks to the Author):

In the manuscript NCOMMS-19-15028, entitled 'An effector protein of the wheat stripe rust fungus targets chloroplasts to suppress chloroplast-mediated host immunity', Xu and colleagues aimed at functionally characterizing the *Puccinia striiformis* f sp *tritici* putative effector Pst_12806. To achieve this goal, the authors combined approaches based on cell biology, protein-protein interaction, gene expression, and in planta assays (HIGS, cell death, callose deposition, ...). They report that Pst_12806 i) is important for *P. striiformis* f sp *tritici* pathogenicity, ii) targets chloroplasts with a transit peptide, iii) interacts with the TaISP host protein, iv) affects chloroplast functions and attenuates photosynthesis, v) suppresses immunity (both PTI and ETI) and H₂O₂ generation, and iv) that TaISP positively regulates photosynthesis. They conclude that Pst_12806 translocates into chloroplasts to alter photosynthesis and avoid triggering cell death.

The manuscript tackles a timely and important question and focuses on an important pathogen. The findings reported would advance, though incrementally, our understanding of plant-microbe interactions. Unfortunately, I have numerous serious concerns about the validity of the data and of the conclusions. Notably, I feel that key conclusions are not supported by the data in the

manuscript, because of critical experimental flaws and the lack of key controls in many experiments. For instance, when performing experiments in planta, the authors should include control proteins that accumulate in the same subcellular compartment than the protein tested. Such controls are systematically missing at the moment. See for instance the suppression of cell death experiments (fig. 2 - need for a secreted protein as a control), suppression of PTI assays (fig. 3 - need for a chloroplast-targeted GFP as a control), or BiFC (fig. 5 - both YFP parts must accumulate in the chloroplast to have a proper control). Also, microscopy data miss key features to controls chlorophyll autofluorescence (see. figs. 5 and 7 for instance - guard cells should be included on images to control chlorophyll autofluorescence) and/or are of too poor quality to conclude regarding protein accumulation patterns. Also, HIGS assays show weak differences (and large/overlapping error bars), that do not convincingly support authors conclusions.

other comments

- Fig 1. I recommend to show lower magnification images to allow readers to evaluate other potential locations of the fusions. Also, the blots show no sign of cTP cleavage. I would expect to see two bands (cleaved and un-cleaved). Could the authors comment on that?
- fig. 2. over-expression of secreted protein often artifactually suppress BAX-induced cell death, due to unfolded protein response because of overaccumulation of proteins in the ER. I recommend to re-do the experiment with the effector construct without the SP.
- fig 4. Bar charts show weak differences and large overlapping error bars. How could the means be significantly different? What is the reliability/validity of statistical tests when n=3?
- Fig S2. I recommend to show the microscopy image of the GFP-effector fusion. It should show no sign of accumulation in the chloroplasts if the cTP is truly functional. Also, the cTP panel should include a guard cell to control chlorophyll autofluorescence. Without that, it is impossible to know that the green signal is truly coming from GFP (and is not an artifact).
- Fig S3. I recommend using higher quality images, and avoid over-cropping blot images.
- Introduction is inaccurate and misses key information in some places. For instance, l. 61: as far as I know, there is no example of NB-LRR proteins that 'inactivate' effectors. They just recognize them and signal this recognition. l. 85. Incorrect reference. MLP124111 was shown to accumulate in chloroplasts in Petre et al., 2015, MPMI. Also, many rust candidate effectors have been shown to accumulate in chloroplasts, some using cTPs, that are not mentioned in the introduction (see Lorrain et al., 2018, Current Opinion in Microbiology).
- Fig S5. I recommend quantifying such data. Are the large bright dots callose...? They look like large dust artifacts ...
- there is no clear evidence that PST_12806 is a secreted protein (l. 26 in the abstract is misleading and an overstatement).
- l. 237 'interaction was verified'. Scientists should never perform experiments to 'verify' preliminary observations; they do so to test/challenge their robustness. If such wording were to reflect the way authors reason, then they would raise serious concerns about the validity of the scientific approach and the validity of the subsequent findings.

Dears,

Thank you for the comments and suggestions about our manuscript entitled 'An effector protein of the wheat stripe rust fungus targets chloroplasts to suppress chloroplast function'. All comments are valuable and very helpful for improving this manuscript. We considered all the comments carefully, have repeated all experiments as suggested. The detailed responses are as follows.

Reviewer #1 (Remarks to the Author):

In their manuscript the authors present an effector protein from the wheat stripe rust that targets chloroplast mediated host immunity. The authors provide ample evidences that the effector target the chloroplast, the experiment all have appropriate control, most of the conclusion are sound and the manuscript is very well presented.

Il together I have only two major issue which I would like better explained in the manuscript, othe rissues are minor.

Major issue 1). In line 227 the authors mention that 95 Y2H clones were recovered, from which two partial TaISP sequences were recovered. Hence the rest of the manuscript focuses on the TaISP. It seems that it needs to be justified why the TaISP was selected and not any other interacting protein from the list of 40 interactants, if not it really looks as cherry picking. In addition (I do not blame the authors for that), but the supplementary data is very hard to analyse, I think the submission portal has separated the columns of Y2H results which appear on separate pages, this would have to be solved prior to publication.

Response: Effector Pst_12806 was predicted to be localized into chloroplasts and we focused on the chloroplast-related interactant in this study (we stated this point on line 246-247, page 10). Among 40 interactants, there are six chloroplast-related interactants and all of them were tested to interact with Pst_12806. Finally, Only TaISP interacted with Pst_12806. The supplementary data 1 also was revised as suggested to adjust the columns.

Major issue 2: the title states ...supress chloroplast-mediated host immunity.
I agree that the data presented in the manuscript clearly demonstrate that chloroplast function and H₂O₂ accumulation are affected. However, these processes are not strictly and only related to host immunity, for this reason I believe the title goes to far. Saying it supresses chloroplast function would be OK, but what is the evidence that immunity specific aspects of the chloroplast are suppressed?

Response: We changed the title into ‘An effector protein of the wheat stripe rust fungus targets chloroplasts to suppress chloroplast function’

Minors

Line 87. In ref 16 expression was in *A. thaliana* not *N benthamiana*

Response: We revised it.

Line 133. compared to that (not to than)

Response: We revised it.

Line 179. Indicate Ethan DsRed in the figure

Response: We revised it.

Line 297. As with the negative...

Response: We revised it.

Line 309 to 318. There are discrepancies between the letters indicated in the text and the labeling of the panels of the figure.

Response: We revised it.

Reviewer #2 (Remarks to the Author):

In the manuscript NCOMMS-19-15028, entitled 'An effector protein of the wheat stripe rust fungus targets chloroplasts to suppress chloroplast-mediated host immunity', Xu and colleagues aimed at functionally characterizing the *Puccinia striiformis* f.sp. *tritici* putative effector Pst_12806. To achieve this goal, the authors combined approaches based on cell biology, protein-protein interaction, gene expression, and in planta assays (HIGS, cell death, callose deposition, ...). They report that Pst_12806 i) is important for *P. striiformis* f.sp. *tritici* pathogenicity, ii) targets chloroplasts with a transit peptide, iii) interacts with the TaISP host protein, iv) affects chloroplasts functions and attenuates photosynthesis, v) suppresses immunity (both PTI and ETI) and H₂O₂ generation, and iv) that TaISP positively regulates photosynthesis. They conclude that Pst_12806 translocates into chloroplasts to alter photosynthesis and avoid triggering cell death.

The manuscript tackles a timely and important question and focuses on an important pathogen. The findings reported would advance, though incrementally, our understanding of plant-microbe interactions. Unfortunately, I have numerous serious concerns about the validity of the data and of the conclusions. Notably, I feel that key conclusions are not supported by the data in the manuscript, because of critical experimental flaws and the lack of key controls in many experiments. For instance, when performing experiments in planta, the authors should include control proteins that accumulate in the same subcellular compartment than the protein tested.

Response: We provided CTP1 protein as a positive control that accumulate in chloroplast (page 6, line 154-157). CTP1 protein from *Melampsora larici-populina* acted as a chloroplasts-targeted protein in Petre et al., 2015, *Cellular microbiology*. Pst_12806:GFP and CTP1:CFP were co-expressed in tobacco cells and the result indicated that Pst_12806 was localized into chloroplast.

Such controls are systematically missing at the moment. See for instance the suppression of cell death experiments (fig. 2 - need for a secreted protein as a control

Response: we provided a secreted protein Pst_01425 as a negative control which could not suppress Bax-induced cell death and the secreted protein Avr1b as a positive control. (page 7-8, line 187-189).

suppression of PTI assays (fig. 3 - need for a chloroplast-targeted GFP as a control

Response: We provided cTP:GFP as a negative control in PTI assays and we also repeated the PTI assay. cTP:GFP could accumulated into chloroplast in this research (Supplementary Fig. 3).

or BiFC (fig. 5 - both YFP parts must accumulate in the chloroplast to have a proper control).

Response: Thank you for your comments, we repeated the BIFC assay and provided two groups of negative controls (one group is Pst_12806- cEYFP and cTP-nEYFP, the other is cTP-cEYFP and TaISP-nEYFP; see Fig.5). The cTP of Pst_12806 could carry GFP to translocated into chloroplasts (Supplementary Fig. 3).

Also, microscopy data miss key features to controls chlorophyll autofluorescence (see. figs. 5 and 7 for instance - guard cells should be included on images to control chlorophyll autofluorescence) and/or are of too poor quality to conclude regarding protein accumulation patterns.

Response: In Fig.5, we repeated the BIFC assay and provided large better field images and bright field images (see clearly guard cells) for BIFC interaction. we did not see any fluorescence in guard cell in Fig.5, indicating that the signal was not from chlorophyll autofluorescence. In Fig.7, we also repeated this assay and provided a better image and bright field images (see clearly guard cells). The fluorescence image was the oxidized dichlorofluorescein (DCF) signal of H₂O₂ production in chloroplast, other organelles and cytoplasmic membrane. There is no fluorescence in guard cell in Fig.7, thus, the fluorescence in chloroplasts was not from chlorophyll autofluorescence.

Also, HIGS assays show weak differences (and large/overlapping error bars), that do not convincingly support authors conclusions.

Response: We repeated HIGS assay. Means and standard deviation were calculated from six biological replicates. Statistical analysis showed that the differences were significant. We also observed HR (hypersensitive response), and the accumulation of H₂O₂ was significantly increased in *Pst_12806*-silenced plants (see Fig.4b and Fig7d).

The development of *Pst* (hyphae length and infected area) was compromised compared with the control in *Pst_12806*-silenced plants (Fig.4e).

other comments

- Fig 1. I recommend to show lower magnification images to allow readers to evaluate other potential locations of the fusions. Also, the blots show no sign of cTP cleavage. I would expect to see two bands (cleaved and un-cleaved). Could the authors comment on that?

Response: Thank you for your comments, we have relocated a large field of image in Fig.1, and we confirmed that *Pst_12806* accumulated into chloroplasts. It is difficult to observe clearly the cleaved and un-cleaved band with the naked eye. We observed the band of *Pst_12806*: GFP was smaller than the band of GFP: *Pst_12806* in Supplementary Fig. 2b, indicating that *Pst_12806* may be cleaved in plant cell. Similarly, CTP1 protein from *Melampsora larici-populina* acted as a chloroplasts-targeted protein in this paper also had one band in western blot. This is the same case in Petre et al., 2015, Cellular microbiology. But CTP2 appeared obviously cleaved and un-cleaved bands (Petre et al., 2015, Cellular microbiology). The cleaved and un-cleaved bands were present due to different cutting efficiency.

- fig. 2. over-expression of secreted protein often artifactually suppress BAX-induced cell death, due to unfolded protein response because of overaccumulation of proteins in the ER. I recommend to re-do the experiment with the effector construct without the SP.

Response: Thank you for your suggestion. We repeated this experiment with the effector without the SP (page 7, line 187-189; See Fig. 2). *Pst_12806*^{ΔSP} also suppressed Bax-induced PCD in *N. benthamiana*.

- fig 4. Bar charts show weak differences and large overlapping error bars. How could the means be significantly different? What is the reliability/validity of statistical tests when n=3?

Response: Thank you for your comments. The significantly difference were assessed using Unpaired two-tailed Student's t-test. We also repeated this experiment, and mean and standard deviation of hyphal length and infection area were calculated from six biological replicates. 50 infection sites were calculated in every biological replicate.

- Fig S2. I recommend to show the microscopy image of the GFP-effector fusion. It should show no sign of accumulation in the chloroplasts if the cTP is truly functional. Also, the cTP panel should include a guard cell to control chlorophyll autofluorescence. Without that, it is impossible to know that the green signal is truly coming from GFP (and is not an artifact).

Response: Thank you for your suggestion. We originally observed the microscopy image of the GFP-Pst_12806. The signals of GFP-Pst_12806 accumulated mainly in the cytoplasm and nucleus instead of chloroplasts. We have shown the microscopy image of the GFP- Pst_12806 in Supplementary Fig. 2a. The large image also included guard cell to control chlorophyll autofluorescence as suggested. which indicated that the GFP signal is truly coming from cTP:GFP instead of chlorophyll autofluorescence.

- Fig S3. I recommend using higher quality images, and avoid over-cropping blot images.

Response: Figure S3 was revised as suggested to use images with better quality.

- Introduction is inaccurate and misses key information in some places. For instance, l. 61: as far as I know, there is no example of NB-LRR proteins that 'inactivate' effectors. They just recognize them and signal this recognition.

Response: We have changed 'inactivate' into 'recognize'.

l. 85. Incorrect reference. MLP124111 was shown to accumulate in chloroplasts in Petre et al., 2015, MPMI. Also, many rust candidate effectors have been shown to accumulate in chloroplasts, some using cTPs, that are not mentioned in the introduction (see Lorrain et al., 2018, Current Opinion in Microbiology).

Response: We have revised the reference ‘Petre et al., 2015, MPMI’. We did not detailly mention other rust effectors in chloroplasts in introduction part. But this we stated some chloroplast-targeted effectors from *Puccinia* species on line 354-366, page 13.

line 354-366, page 13: The presence of these chloroplast-targeting effectors in the genome suggests that *Pst* and other *Puccinia* species could hijack the plant transport system to sort into chloroplasts and interfere with chloroplast functions as one of the strategies to suppress plant immunity. Several effectors targeting chloroplasts have been identified in *Puccinia* species by LOCALIZER prediction⁴⁵ or subcellular location. When expressed in *N. benthamiana*, two *Pst* effectors (Pst_03196 and Pst_18220) and two *Pgt* effectors (PGTG00164 and PGTG06076) were translocated into chloroplasts^{45, 46, 47}. In another rust fungus *Melampsora larici-populina*, CTP1 was shown to enter the stroma of chloroplasts with the help of its transit peptide²⁹.

- Fig S5. I recommend quantifying such data. Are the large bright dots callose...? They look like large dust artifacts.

Response: We have quantified this data in Supplementary Fig. 6b and we also have provided higher quality image in Supplementary Fig. 6a.

- there is no clear evidence that PST_12806 is a secreted protein (l. 26 in the abstract is misleading and an overstatement).

Response: We have confirmed the secretion function of the signal peptide of Pst_12806 in yeast system and the yeast system had confirmed to identified the secretion function of the signal peptide (Jacobs KA et al., 1979, Gene; Sang-Keun Oh et al., 2009, Plant Cell). The detail result on line 139-150, page 6 and we also provided the related images in Supplementary Fig. 2.

line 139-150, page 6: To verify the function of the signal peptide of Pst_12806, Pst_12806SP (the signal peptide of Pst_12806), Pst_12806^{ΔSP} and Avr1bSP (the signal peptide of Avr1b) were introduced into pSuc2t7M13ori and transformed into yeast strain YTK12 lacking a secreted invertase. Trans-formants were streaked on CMD-W plates and YPRAA plates which only support the growth of yeast with secreting invertase²⁷. Like the positive Avr1bSP, the transformant containing Pst_12806SP grew on the YPRAA plates, but the transformant containing

Pst_12806 Δ SP and the empty vector failed to grow on the YPRAA plates (Supplementary Fig. 2a). The enzyme activity of secreted invertase was also detected by the reduction of 2, 3, 5-triphenyltetrazolium chloride (TTC) and the secreted invertase of the transformant containing Avr1bSP and Pst_12806SP were detected by TTC assay, but the transformant containing Pst_12806 Δ SP and the empty vector not (Supplementary Fig. 2b). These results support the functionality of the signal peptide of Pst_12806.

- l. 237 'interaction was verified'. Scientists should never perform experiments to 'verify' preliminary observations; they do so to test/challenge their robustness. If such wording were to reflect the way authors reason, then they would raise serious concerns about the validity of the scientific approach and the validity of the subsequent findings.

Response: We have changed 'verified' into 'tested'

REVIEWERS' COMMENTS:

Reviewer #2 (Remarks to the Author):

In the manuscript NCOMMS-19-15028A, entitled 'An effector protein of the wheat stripe rust fungus targets chloroplasts to suppress chloroplast function', Xu and colleagues present a revised version of the manuscript NCOMMS-19-15028; manuscript for which I wrote a review report in June 2019. In the revised version, the authors satisfactorily addressed the concerns I raised in my previous review report. I also appreciate the way the authors engaged in a constructive scientific discussion in the rebuttal letter. I am globally positive about the manuscript and I am happy to recommend the manuscript for publication.

Should the authors still wish to improve the manuscript, I suggest paying attention to the two minor comments below. I hope the authors find these useful.

Ben Petre

--

First, the manuscript may require proof reading and editing to correct over- and inaccurate statements. As an example of inaccurate statement, consider the third sentence of the abstract '... that was secreted and translocated into chloroplasts after the cleavage at the predicted transit peptide.' This is inaccurate, as the cleavage occurs after translocation in chloroplasts, not before. As an example of over-statement, consider the sixth sentence of the abstract '...and development of Pst were compromised in Pst_12806 knockdown plants by HIGS...'. The verb compromised seems too strong when looking at the data in figure 4; 'limited' or 'reduced' seems more appropriate to me.

Second, I am still not fully convinced by the differences shown in Figure 4 (especially in the charts reporting hyphae length and infection area), especially since statistical tests made with a limited number of replicates are not particularly meaningful. Should the authors wish to make the graphs more convincing to the reader, I suggest using categorical scatterplots (see the following link for more information -

<https://journals.plos.org/plosbiology/article?id=10.1371/journal.pbio.1002128>), so that readers can more accurately appreciate the distribution of the values for each replicate.

Dears,

Thank you for the comments and suggestions about our manuscript entitled 'An effector protein of the wheat stripe rust fungus targets chloroplasts and suppresses chloroplast function'. All comments are valuable and very helpful for improving this manuscript. We considered the comments carefully and the detailed responses are as follows.

REVIEWERS' COMMENTS:

Reviewer #2 (Remarks to the Author):

In the manuscript NCOMMS-19-15028A, entitled 'An effector protein of the wheat stripe rust fungus targets chloroplasts to suppress chloroplast function', Xu and colleagues present a revised version of the manuscript NCOMMS-19-15028; manuscript for which I wrote a review report in June 2019. In the revised version, the authors satisfactorily addressed the concerns I raised in my previous review report. I also appreciate the way the authors engaged in a constructive scientific discussion in the rebuttal letter. I am globally positive about the manuscript and I am happy to recommend the manuscript for publication.

Should the authors still wish to improve the manuscript, I suggest paying attention to the two minor comments below. I hope the authors find these useful.

Ben Petre

First, the manuscript may require proof reading and editing to correct over- and inaccurate statements. As an example of inaccurate statement, consider the third sentence of the abstract '... that was secreted and translocated into chloroplasts after the cleavage at the predicted transit peptide.' This is inaccurate, as the cleavage occurs after translocation in chloroplasts, not before.

Response: Thank you for your suggestions and I am sorry for the inaccurate statements. And we revised the inaccurate statements and we also agree the editor's edit. The statement is: 'Here, we identified a haustorium-specific protein (Pst_12806) from the wheat stripe rust fungus, *Puccinia striiformis* f. sp. *tritici* (*Pst*), that is translocated into chloroplasts and affects chloroplast function.'

As an example of over-statement, consider the sixth sentence of the abstract '...and development of Pst were compromised in Pst_12806 knockdown plants by HIGS...'. The verb compromised seems too strong when looking at the data in figure 4; 'limited' or 'reduced' seems more appropriate to me.

Response: Thank you for your suggestions. We revised the verb 'compromised' into 'reduces'.

Second, I am still not fully convinced by the differences shown in Figure 4 (especially in the charts reporting hyphae length and infection area), especially since statistical tests made with a limited number of replicates are not particularly meaningful. Should the authors wish to make the graphs more convincing to the reader, I suggest using categorical scatterplots (see the following link for more information - <https://journals.plos.org/plosbiology/article?id=10.1371/journal.pbio.1002128>), so that readers can more accurately appreciate the distribution of the values for each replicate.

Response: Thank you for your suggestion. Hyphae length and infection area were presented by categorical scatterplots in Figure 4.